# Disturbance-Observer-Based U-Control (DOBUC) for Nonlinear Dynamic Systems

**DOI:** 10.3390/e23121625

**Published:** 2021-12-02

**Authors:** Ruobing Li, Quanmin Zhu, Jun Yang, Pritesh Narayan, Xicai Yue

**Affiliations:** 1Department of Engineering Design and Mathematics, University of the West of England, Frenchay Campus, Coldharbour Lane, Bristol BS16 1QY, UK; quan.zhu@uwe.ac.uk (Q.Z.); pritesh.narayan@uwe.ac.uk (P.N.); alex.yue@uwe.ac.uk (X.Y.); 2Department of Aeronautical and Automotive Engineering, Loughborough University, Leicestershire LE11 3TU, UK; J.Yang3@lboro.ac.uk

**Keywords:** disturbance-observer-based control (DOBC), U-model, U-control, disturbance-observer-based U-control (DOBUC)

## Abstract

U-model, which is a control-oriented model set with the property of generally facilitate nonlinearity dynamic inversion/cancellation, has been introduced to the Disturbance Observer-Based control (DOBC) methods to improve the performance of the nonlinear systems in this paper. A general DOB based U-Control (DOBUC) framework is proposed to improve the disturbance attenuation capability of U-controller for both linear and nonlinear systems combined with (based on) the U-model-based dynamic inversion which expands the classical linear disturbance observer control to general nonlinear systems. The proposed two-step DOBUC design procedures in which the design of DOB and U-controller are totally independent and separated, enables the establishment of global exponential stability without being subject to disturbances and uncertainties. Comparative simulation experiments with Nonlinear DOBC in controlling Wind Energy Conversion Systems (WECS) and Permanent Magnet Synchronous Motors (PMSM) demonstrated the proposed method.

## 1. Introduction

It is believed that unknown system uncertainties and disturbances universally exist in various engineering control systems [1]. These uncertainties have critical side effects on practical engineering system operations and academic research. The issue of disturbance rejection is an ongoing topic since the appearance of control theory and applications, among which Disturbance-Observer-Based Control (DOBC) has showed its efficiency because of its potential feature of the “separation principle” for the ease of control design [1,2,3]. In DOBC systems, there are two separate design control loops: a baseline feedback controller to satisfy the desired tracking performance and a Disturbance Observer (DOB) to estimate the system disturbances and uncertainties. When the control system matched perfectly, the DOB is not activated, and consequently, the DOBC reduces to a feedback control system [1]. Different from the worst-case robust control methods which need to sacrifice control performance, the DOBC can preserve the nominal performance to achieve better robustness.

The method to estimate system uncertainties and disturbances accurately is the core element in DOBC design. The fundamental idea of the DOB is to integrate all the external disturbances and internal uncertainties into single lumped errors. The basic DOB structure was first proposed in frequency-domain in the 1980s by Ohishi [4], which estimates the disturbance by using the difference between the system control input and the calculation input obtained through the inversion of the controlled plant model. It should be noted that frequency-domain DOB requires the linear control systems, or ignore the nonlinear parts in nonlinear systems and regard them as disturbances. However, for most control systems, nonlinear dynamic modelling is achievable and its parameters are measurable [1], the estimation performance by linear DOB approach is much limited, so that, instead of broadly assuming as unknown elements, reasonably using this knowledge could facilitate more effective controller designs. Accordingly, the estimation and attenuation of disturbances and uncertainties can be remarkably improved by utilizing the recognized nonlinear dynamics in the design.

Chen [5] firstly proposed a universal nonlinear DOB (NDOB) for disturbance torques estimation caused by unknown friction in nonlinear robotic manipulators, then Chen [2] used Lyapunov stability theory to illustrate the estimation error of disturbances will converge exponentially to 0 eventually. The NDOB described in [5] was investigated further in [6], where the control performance of a missile autopilot system had been improved. In addition, NDOB approaches have been widely implanted to many real-time applications, such as the roll, attack and sideslip angle control design for UAV [7], the position control of a seven degrees of freedom manipulator [8], and roll tracking control design of Quadrotor [9]. It may be seen that the previous discussed NDOB approaches are proposed in time-domain, with the disturbance estimation in each system state variables’ channel. Compared with time-domain DOB, frequency-domain DOB has its unique advantage: can integrate all the external system disturbances and internal modelling uncertainties into control input error, which is more intuitive, less computation and its adjustment object is only the design of the low-pass filter [10]. However, because frequency-domain DOB requires the inversion of the plant model, which is difficult to obtain for a nonlinear system, it cannot be directly applicable to nonlinear systems. Therefore, if one can construct the inverse dynamic of nonlinear systems, the lumped disturbance can then be estimated directly using the difference between system input and calculated input.

Another aspect of DOBC is the baseline controller designed to satisfy the required system performance specifications. U-model based control (U-control) proposed and developed by [11,12], takes efficiency of UM dynamic inversion and can applied as a proper control strategy. By getting an inversion, the controlled plant can be compensated into “1” (a unit constant), which brings a designer’s requested system response and phase delay-free between system inputs and outputs. Accordingly, the superiority of U-control has attracted a number of research initiatives. For example, U-Pole Placement Control (UPPC) [13], U-General Predictive Control (UGPC) [14], U-Neuro-Control (UNC [15], U-Internal Model Control (UIMC) [16], and U-Two-Degree-of-Freedom IMC [17]. However, there are critical issues in dynamic inversion before making any designed U-control system applicable in real situations. Because the inverter of U-controller is highly sensitive and dependent on the credibility/accuracy of the controlled plants, most existing U-control approaches require accurate modelling without uncertainties. Accordingly, U-control must consider such robustness in its designed control systems. It should be noted that almost all U-control is still focused on its early research—model matched control, that is, with the assumption of model known exactly, then put the focus on those fundamental structural issues.

Motivated by the advantages of the U-control and the frequency-domain DOB, this paper aims to develop a nonlinear DOB design, which is analogous to the conventional frequency-domain DOB. Promisingly, as described in U-Model (UM) based dynamic inversion algorithm [18], the inverse of the nonlinear system has been used to design a part of the U-controller. Based on the simulation experiments, it also verifies the efficiency of UM based dynamic inversion algorithm can be applied into not only affine nonlinear system, but also non-affine nonlinear system. This offers the possibility of extending the idea of frequency-domain DOB into a new disturbance estimation approach using the difference between system input and calculated input. In summary, there are two main contributions in the paper. The first one is to propose a new nonlinear DOB design method, extend the idea of conventional frequency-domain DOB be applicable to nonlinear system, which can integrate all external disturbances and internal modelling uncertainties as system input error, whereas the conventional time-domain nonlinear DOB estimates the disturbance in system state variables’ channel. The second one is combining the strengths exhibited in U-control and this new nonlinear DOB to provide an enhanced, fast-response, delay-free, and convenient design control framework, applicable to all linear/nonlinear invertible controlled plants. The remainder of this paper is organized as follows: Section 2 states the excited issues of nonlinear system control using DOB methods. Section 3 proposes DOBUC design framework and performance analysis. Section 4 takes simulation experiments to illustrate and demonstrate the findings of the studies. Finally. conclusions are presented in Section 5.

## 2. Issues of Nonlinear System Control Using DOB

### 2.1. Frequency Domain DOBC

Ohishi [4] firstly proposed the basic diagram (shown in Figure 1) of frequency-domain DOB by ignoring the outer feedback loop, which is implementable for minimum-phase systems. As described in [10], it is extended to nonminimum-phase systems. However, both frequency-domain DOB requires linear systems, or their nonlinear properties must be estimated as disturbance variables. Figure 1 shows the basic structure of frequency-domain DOB control system.

Where $G_{P}\left(s\right)$ is the actual physical plant, $G_{0}\left(s\right)$ is its nominal model, r is the reference signal, d is the external system disturbance, $e_{u}$ is the lumped disturbance, and ${\hat{e}}_{u}$ is the estimate of the lumped disturbance. From Figure 1, the baseline controller C(s) in outer loop is designed according to desired tracking performance and the observer in inner loop is designed to reject disturbance and suppress uncertainty. The tracking and robustness requirements can be implemented by separately designing of the normal feedback loop and disturbance compensation loop. It should be noted that when system modelling without disturbance and uncertainty, the disturbance estimation and compensation module in the inner loop will not be activated. Consider a SISO linear minimum phase system by:(1)$$y=G_{P}\left(s\right)\left[u+d\right]$$
where u is the ideal control system input, $u_{c}$ and y are the controller and the control system output, respectively. It should be clarified that both external system disturbances caused by noise and internal disturbances caused by modelling uncertainties can be estimated by frequency-domain DOB. Equation (2) indicates the lumped disturbance $e_{u}$ in Figure 1 contains two items as
(2)$$\begin{array}{l} e_{u}=yG_{0}{\left(s\right)}^{-1}-u\\ =G_{P}\left(s\right)\left[u+d\right]G_{0}{\left(s\right)}^{-1}-u\\ =G_{P}\left(s\right)G_{0}{\left(s\right)}^{-1}\left[u+d\right]-G_{P}\left(s\right)G_{P}{\left(s\right)}^{-1}\left[u+d\right]+d\\ =\left[G_{P}{\left(s\right)}^{-1}-G_{0}{\left(s\right)}^{-1}\right]G_{P}\left(s\right)\left[u+d\right]+d\\ =\left[G_{P}{\left(s\right)}^{-1}-G_{0}{\left(s\right)}^{-1}\right]y+d \end{array}$$

The first item in (2) relates to the modelling errors between the controlled $G_{P}\left(s\right)$ and the nominal model $G_{0}\left(s\right)$, the second relates to the external disturbance d. Therefore, $e_{u}$ contains all of the disturbances and uncertainties. After letting it pass a filter Q(s), the lumped disturbance estimation ${\hat{e}}_{u}$ is:(3)$${\hat{e}}_{u}=Q\left(s\right)G_{0}{\left(s\right)}^{-1}y-Q\left(s\right)u=Q\left(s\right)e_{u}\;$$

The lumped disturbance estimation error, that is, ${\hat{e}}_{u}-e_{u}$ will need to be zero when time goes to infinity. Clearly, filter Q(s) should be selected as a low-pass filter, that is, in the frequency range of Q(jω)=1. It is derived that the control system output is:(4)$$y=G_{P}\left(s\right)\left[\frac{G_{0}\left(s\right)}{G_{0}\left(s\right)+Q\left(s\right)\left[G_{P}\left(s\right)-G_{0}\left(s\right)\right]}u_{c}-\frac{G_{P}\left(s\right)Q\left(s\right)}{G_{0}\left(s\right)+Q\left(s\right)\left[G_{P}\left(s\right)-G_{0}\left(s\right)\right]}d+d\right]\;=\frac{G_{0}\left(s\right)G_{P}\left(s\right)}{G_{0}\left(s\right)+Q\left(s\right)\left[G_{P}\left(s\right)-G_{0}\left(s\right)\right]}u_{c}+\frac{G_{P}\left(s\right)G_{0}\left(s\right)\left[1-Q\left(s\right)\right]}{G_{0}\left(s\right)+Q\left(s\right)\left[G_{P}\left(s\right)-G_{0}\left(s\right)\right]}d$$

Clearly, filter Q(s) should be selected as a low-pass filter, that is, in the frequency range of Q(jω)=1, it follows from (4) that:(5)$$\underset{\omega \to 0}{\lim}y=G_{0}\left(j\omega \right)u_{c}+0d$$

Equation (5) implies that the disturbances and uncertainties in system have been eliminated by the frequency-domain DOB.

**Remark** **1.**
*The difference order degree of LPF*

Q(s)

*between the denominator and the numerator should be larger than that of the nominal model*

$G_{0}\left(s\right)$

*to ensure that*

$Q\left(s\right)G_{0}{\left(s\right)}^{-1}$

*be proper.*


**Remark** **2.***Based on (3) Design the LPF*Q(s)*is close to 1 in all frequency range to guarantee accurate estimation of the lumped disturbance that is,*$e_{u}={\hat{e}}_{u}$.

### 2.2. Nonlinear DOB

DOBC for linear system has been developed and employed in engineering over three decades. Ohishi [4] pioneered the development of DOBC for motion control systems. After that, DOBC has been employed in many mechatronic systems including disk drivers, machining centers, dc/ac motors, and manipulators. However, this observer may make the DOB not implementable due to the requirement of the inverse of the nominal plant $G_{P}{}^{-1}$, especially for nonlinear plant [1]. To deal with nonlinear plants and uncertainty, one of the famous solutions is developing DOB into nonlinear DOB (NDOB). 

Consider a classical nonlinear system:(6)$$\{\begin{array}{l} \dot{x}=f\left(x\right)+g_{1}\left(x\right)u+g_{2}\left(x\right)d\\ y=h\left(x\right)\;\;\;\;\;\;\;\;\;\;\;\;\;\;\;\;\;\;\;\;\;\;\;\;\;\;\;\;\;\;\;\;\;\;\;\;\; \end{array}$$
where $x\in R^{n},d\in R^{n},u\in R$ and y∈R are the state vector, the lumped disturbance vector, and the system input and output. It is assumed that f(x), $g_{1}\left(x\right)$, $g_{2}\left(x\right)$, and h(x) are smooth functions in terms of x. Ref. [1] proposed a NDOB to estimate the disturbance for system (6) as:(7)$$\{\begin{array}{l} \dot{z}=-l\left(x\right)g_{2}\left(x\right)z-l\left(x\right)\left[g_{2}\left(x\right)p\left(x\right)+f\left(x\right)+g_{1}\left(x\right)u\right]\\ \hat{d}=z+p\left(x\right)\;\;\;\;\;\;\;\;\;\;\;\;\;\;\;\;\;\;\;\;\;\;\;\;\;\;\;\;\;\;\;\;\;\;\;\;\;\;\;\;\;\;\;\;\;\;\;\;\;\;\;\;\;\;\;\;\;\;\;\;\;\;\;\;\;\;\;\;\;\;\;\;\;\;\;\;\;\; \end{array}$$
where $\hat{d}\in R^{n}$ is an estimate of all the disturbances and uncertainties, $z\in R^{n}$ is the estimates of the internal states of nonlinear observer. Although NDOBC has been successfully applied in a wide range of fields such as industrial control [19] and aircraft control [9], it cannot be denied that its design is complicated [20]. Where l(x), p(x) are all parameters or functions that need to be designed according to the system and need to be stable and satisfied by [2]:(8)$$l\left(x\right)=\frac{\partial p\left(x\right)}{\partial x}\;and\;{\dot{e}}_{d}=-l\left(x\right)g_{2}\left(x\right)e_{d}$$
where $e_{d}=d-\hat{d}$ is the disturbance estimation error. Concretely, NDOBC realizes the efficient control to nonlinear plants by extending the linear disturbance observer to nonlinear disturbance observer. However, time-domain NDOB proposed by [2] is still defective in the following aspects:
(1)The design in time-domain NDOB [2] is more complex than frequency-domain DOB, and therefore the design process of mentioned NDOB is more complicated and requires more expensive computation [20].(2)The hypothesis of NDOB: the lumped disturbance d changes slowly with time [20], because of the limitation of the convergence speed of NDOB; from [20], the state variables of the plant’s model are known or can be directly measured.(3)The lumped disturbances estimated requires the dynamics of disturbance observer be faster than that of the closed-loop dynamics, which may put pressure on industrial design.(4)The estimation results from NDOB cannot be used for compensation directly because the disturbances and control inputs are not in the same channels, where $d\in R^{n}$ but u∈R.

## 3. Disturbance Observer Based U-Control

### 3.1. U-Model Based DOB (UDOB)

From Section 2, both DOB-based control methods have their shortages. Therefore, another interesting and potential solution is considered to only change the inversion method to re-meet the requirements of the frequency-domain DOB of nominal plant $G_{P}$ and expand it to adapt to the inversion of the nonlinear system. Inspired by U-model based (UM) dynamic inversion algorithm [18], which can convert nonlinear inversion into linear expression without losing or ignoring plants’ features or nonlinear functions. Based on linear frequency-domain DOBC structure in Figure 2, combining the advantages of UM dynamic inversion algorithm, a novel UDOB is generated in this paper. Figure 2 shows this structure.

Where $G_{u}$ is the U-relation of nominal plant $G_{P}$ and$\;G_{u}{}^{-1}$ is the U-model based inversion of $G_{u}$, d is the disturbance, $u_{c}$ is the controller output, u and y are control system input and output respectively. UDOB is similar to frequency-domain DOB in structure, and the core of both is the inversion of controlled plant $G_{P}$ and design of the proper low-pass filter Q. The use of UM inversion algorithm expands the inversion of the traditional linear transfer function into an inversion method that can adapt to linear/nonlinear, transfer function/state space/polynomial based plants, and therefore expands its application range on the basis of frequency-domain DOB.

Compared with NDOB, this UDOB has the following advantages:(1)The computation of UDOB is cheaper (only the inversion of controlled plant $G_{u}{}^{-1}$ is required), and there are less parameters to be designed and adjusted (only parameters in the filter Q).(2)The estimates of UDOB can be used to compensate the disturbances directly since the estimation disturbances are in the same channels with the control inputs, where d∈R and u∈R.(3)UDOB does not require the assumption that “the lumped disturbance d changes slowly with time” in [20], because the inversion process happens almost instantaneously if the response of $G_{c1}$ is fast enough and the computation tool is very powerful.

### 3.2. U-Control 

U-control system framework [21] is shown in Figure 3. Assuming the linear/nonlinear controlled plant model $G_{P}$ is stable, bounded, and invertible, the output of U-control system is
(9)$$y=\frac{G_{c1}G_{0}^{-1}G_{P}}{1+G_{c1}G_{0}^{-1}G_{P}}r+\frac{G_{P}}{1+G_{c1}G_{0}^{-1}G_{P}}d$$
where$\;G_{c1}$ is a linear invariant controller, $G_{P}$ is the controlled plant and $G_{0}^{-1}$ is its regular inverter. Model $G_{P}\;\mathrm{equals}\;G_{0}$ achieves when system is free of disturbance and uncertainty, manifestly, the output of U-control system (9) will be simplified to
(10)$$y=\frac{G_{c1}}{1+G_{c1}}r=Gr$$

Concretely, G is the gain of the whole closed-loop control system, which can be designed and adjusted by typically different damping ratio ζ and natural frequency $\omega_{n}$ of $G_{c1}$.

**Remark** **3.**
*For complement of UM dynamic inversion used in U-control system design, the output’s high-order derivatives*

$y^{\left(m\right)}$

*can be obtained by the following methods:*
(1)
*The highest order derivatives*

$y^{\left(M\right)}$

*in (9) can be obtained from the invariant controller*

$G_{c1}$

*[18].*
(2)
*Obtained directly from advanced measuring and gauge equipment. For example, acquire the second derivative of the position signal by measuring the acceleration.*
(3)
*Reshape the polynomial-based model to a chained state space-based model, and the use the state observer to get the value of the desired signal indirectly.*
(4)
*By using high-order filters, such as*

$F=\frac{s^{M}}{{\left(\alpha s+1\right)}^{T}},\;T\geq M$

*, to obtain the required high-order differential signals directly from the system output indirectly.*



**Remark** **4.**
*After the controlled plant is compensated by its inverter, its control characteristics will be offset. Consequently, no amplitude attenuation and phase delay appeal in U-control system ideally.*


### 3.3. DOB Based U-Control (DOBUC)

Although U-control shows high efficiency in the case of model matching, its high sensitivity to inversion [21] limits its real-time control application. UDOB stated its advantages in observing lumped disturbance in Section 3.1, by combining the lumped disturbance and system input into the same channel, the control system design and disturbance compensation will be much easily realized. Accordingly, U-control can cooperate with UDOB’s rapid response, and UDOB can reduce U-control’s sensitivity to accurate modelling of the controlled plants (improving system robustness). Therefore, combining the advantages of both, this paper proposes a novel UDOB enhanced U-control method (DOBUC). The structure of DOBUC is shown in Figure 4.

From Figure 4, this DOBUC has two loops inside and outside. UDOB of inner loop can estimate the lumped disturbances includes the system input disturbances and modelling errors due to uncertainties, U-controller of outer loop will provide the system tracking performance specifications required by user. Manifestly, if control system is perfect matched and without disturbances, the inner UDOB has no contribute to the system. In this case, the system tracking preference can be designed directly through U-control method. When system has modelling error or disturbance, the inner loop is activated and will reject the disturbance and suppress uncertainty. Substantiated in Section 3.1, the estimate of the lumped disturbance in UDOB can be used directly for compensation in the input channel of the U-control system. Demonstrated in Section 3.2, when the inversion of the controlled plant is realized, a proper low-pass filter will force the lumped disturbances be suppressed eventually.

Based on the structure of DOBUC in Figure 4, the U-controller output is:(11)$$u_{c}=\left(r-y\right)G_{c1}G_{u}^{-1}$$

The control system input can be calculated by:(12)$$u_{c}G_{P}-\left(yG_{u}^{-1}-u\right)QG_{P}+dG_{P}=y$$
where $y=\left(u+d\right)G_{P}$ from Figure 4. Then it derived from (12) that:(13)$$u_{c}G_{P}-\left(\left(\left(u+d\right)G_{P}\right)G_{u}^{-1}-u\right)QG_{P}+dG_{P}=\left(u+d\right)G_{P}\;u_{c}G_{P}+dG_{P}-dG_{P}\left(1+G_{u}^{-1}QG_{P}\right)=uG_{P}\left(1+G_{u}^{-1}QG_{P}-Q\right)$$
(14)$$u=\frac{G_{u}}{G_{u}+Q\left(G_{P}-G_{u}\right)}u_{c}-\frac{G_{P}Q}{G_{u}+Q\left(G_{P}-G_{u}\right)}d$$

Therefore, the DOBUC system output can be calculated by:(15)$$\begin{array}{l} y=G_{P}\left(u+d\right)\\ =G_{P}\left[\frac{G_{u}}{G_{u}+Q\left(G_{P}-G_{u}\right)}u_{c}-\frac{G_{P}Q}{G_{u}+Q\left(G_{P}-G_{u}\right)}d+d\right]\\ =G_{P}\left[\frac{G_{u}}{G_{u}+Q\left(G_{P}-G_{u}\right)}\left(r-y\right)G_{c1}G_{u}^{-1}-\frac{\left(1-Q\right)G_{u}}{G_{0}+Q\left(G_{P}-G_{u}\right)}d\right]\;=\frac{G_{P}G_{c1}}{G_{u}+Q\left(G_{P}-G_{u}\right)}r\;-\frac{G_{P}G_{c1}}{G_{u}+Q\left(G_{P}-G_{u}\right)}y\;-\frac{\left(1-Q\right)G_{u}G_{P}}{G_{0}+Q\left(G_{P}-G_{u}\right)}d \end{array}$$

Then (15) becomes:(16)$$y\left(\frac{G_{P}G_{c1}}{G_{u}+Q\left(G_{P}-G_{u}\right)}+1\right)=\frac{G_{P}G_{c1}}{G_{u}+Q\left(G_{P}-G_{u}\right)}r-\frac{\left(1-Q\right)G_{u}G_{P}}{G_{0}+Q\left(G_{P}-G_{u}\right)}d$$

It is derived from (16) that the system output y equals:(17)$$y=\frac{G_{P}G_{c1}}{\left(1-Q\right)G_{u}+G_{P}\left(Q+G_{c1}\right)}r-\frac{G_{P}G_{u}\left(1-Q\right)}{\left(1-Q\right)G_{u}+G_{P}\left(Q+G_{c1}\right)}d\;=\frac{G_{P}G_{c1}}{G_{u}+Q\left(G_{P}-G_{u}\right)+G_{P}G_{c1}}r-\frac{G_{P}G_{u}\left[1-Q\right]}{G_{u}+Q\left(G_{P}-G_{u}\right)+G_{P}G_{c1}}d$$

Clearly, if the filter Q(s) is selected as a low-pass form, that is, $\underset{\omega \to 0}{\lim}Q\left(j\omega \right)=1$, it follows from (17) that:(18)$$\underset{\omega \to 0}{\lim}y=\frac{G_{c1}}{1+G_{c1}}r+0d$$

From (18), modelling errors between $G_{P}$ and $G_{u}$ will not affect system performance and the external disturbances will eliminated by the UDOB, then the structure of DOBUC is equivalent to the structure presented in Figure 5.

In summary, the design procedures of this DOBUC are:(1)Convert the controlled plant GP into its U-model based expression Gu, then design its dynamic inverter Gu−1 through UM dynamic inversion algorithm. Accordingly, the model inverse Gu−1 should exist and satisfy the Lipschitz continuity with globally uniformly:‖G(x1)−G(x2)‖≤γ1G‖x1−x2‖, ∀x1,x2∈ℝn‖G−1(x1)−G−1(x2)‖≤γ2G−1‖x1−x2‖, ∀x1,x2∈ℝn(2)Based on the dynamic inverse in (1) to design a disturbance observer with a suitable low-pass filter Q=1(1+λs)ρ. Parameter ρ should be selected large enough to ensure QGu−1 be proper; λ is the filter time constant, which has an inverse relationship with the speed of closed loop response. It is noted that the smaller the value of λ, the higher accuracy in disturbance estimation.(3)Design invariant controller Gc1 with user-desired damping ratio and undamped natural frequency ωn. Where Gc1=G1−G and G=ωn2s2+2ωns+ωn2.

**Remark** **5.**
*UDOB can be combined with any other baseline control method, because it can integrate the lumped disturbances and system input in the same channel. The use of U-control method here is not only because of its superior control tracking performance, but also because it is very convenient and elementary to design U-controller based on the realization of controlled plants’ inversion.*


**Remark** **6****.**
*Because of the restriction of inversion, UDOB is not effective enough for non-minimum phase systems, while time-domain DOB does not have this restriction [20]. Therefore, although UDOB shows its superiority, it is still necessary to select a suitable disturbance observer according to the controlled plants in practice.*


## 4. Simulation Experiments

### 4.1. Control of Wind Energy Conversion System (WECS)

Compared with traditional non-renewable energy sources, such as petroleum in fossil fuels, environmentally friendly wind energy is widely concerned by the industrial communities because of its wide distribution and no greenhouse gas generation [22]. In a WECS, wind energy is converted into electrical energy requiring wind turbines. Variable-speed wind turbines (VSWT) have received widespread attention due to their potential to obtain maximum power generation [23]. Therefore, an efficient VSWT control strategy is essential to reduce high operating costs and improve power quality [24]. There are two main challenges in WECS [25]: complex nonlinear dynamics and difficult to measure actual wind speed in real-time application [26,27,28] collect the latest research on advanced control and optimization examples of wind energy systems [18] first proposed a continuous-time U-control system for wind energy conversion. Although it does not consider unmeasurable wind speed disturbance and modelling uncertainties, the processing of nonlinear dynamics and excellent power tracking performance under perfect matching conditions are still worthy for further study. Therefore, this paper introduces UDOB and try to solve the weakness of U-control in the design of WECS.

#### 4.1.1. Modelling of WECS 

The modeling of wind turbines plays an important role in the development of an efficiency control strategies for the optimal operation [25]. This section characterizes the wind turbine model as an integrated nonlinear dynamic plant operation model to describe the input/output relationship.

Inspired by the model of WECS in [25], the system output aerodynamic torque can be described by:(19)$$T_{a}=\frac{1}{2}\rho \pi R^{3}C_{q}\left(\lambda ,\beta \right){\left(\hat{v}-\xi \right)}^{2}$$
where ρ and R are the air density and rotor radius; $\hat{v}$ and ξ are the effective wind speed estimate and speed measurement noise; $C_{q}\left(\lambda ,\beta \right)$ is the wind turbine power conversion efficiency, which depends on the tip-speed ratio λ and blade pitch angle β is given as:(20)$$C_{q}\left(\lambda ,\beta \right)=\frac{0.22}{\lambda }\left(\frac{116}{m}-0.4\beta -5\right)\exp\left(-12.5/m\right)$$
where
(21)$$\frac{1}{m}=\frac{1}{\lambda +0.08\beta }-\frac{0.035}{\beta^{3}+1}$$

The tip-speed ratio λ is defined as:(22)$$\lambda =\frac{R\omega_{r}}{v}$$
where $\omega_{r}$ is the rotor angular speed. Therefore, the rotor power $P_{a}$ is generated from (22) as:(23)$$P_{a}=\omega_{r}T_{a}=\frac{1}{2}\rho \pi R^{2}C_{p}\left(\lambda ,\beta \right){\left(\hat{v}-\xi \right)}^{3}$$
where
(24)$$C_{p}\left(\lambda ,\beta \right)=C_{q}\left(\lambda ,\beta \right)\lambda$$

From the conversion system drive train scheme shown in Figure 6, the rotor speed $\omega_{r}$ is driven by the rotor torque $T_{a}$ and the low-speed torque $T_{ls}$. The generator speed $\omega_{g}$ is driven by the high-speed torque $T_{hs}$ and the electromagnetic torque$\;T_{em}$. The generator can get high speed from the rotor through the gearbox, their dynamics can be defined as:(25)$$\{\begin{array}{c} J_{r}{\dot{\omega }}_{r}=T_{a}-K_{r}\omega_{r}-T_{ls}\\ J_{g}{\dot{\omega }}_{g}=T_{hs}-K_{g}\omega_{g}-T_{em} \end{array}$$
where $J_{r}$ and $K_{r}$ are rotor inertia and external damping, $J_{g}$ and $K_{g}$ are generator inertia and external damping. The conversion rate between $\omega_{r}$ and $\omega_{g}$ can be described by gearbox ratio $n_{g}$ as
(26)$$n_{g}=\frac{\omega_{g}}{\omega_{r}}=\frac{T_{ls}}{T_{hs}}$$

Invoking (26), the generator dynamic in (25) can be rewritten as
(27)$$n_{g}{}^{2}J_{g}{\dot{\omega }}_{r}=T_{ls}-n_{g}{}^{2}K_{g}\omega_{r}-n_{g}T_{em}$$

Thus, the drive train model can be described by combining (25) with (27) as
(28)$$J_{t}{\dot{\omega }}_{r}=T_{a}-K_{t}\omega_{r}-T_{g}$$
where
(29)$$\{\begin{array}{c} J_{t}=J_{r}+n_{g}^{2}J_{g}\\ K_{t}=K_{r}+n_{g}^{2}K_{g}\\ T_{g}=n_{g}T_{em} \end{array}$$

Then, the generator output power can be given as:(30)$$P_{g}=T_{g}\omega_{r}$$

#### 4.1.2. DOBUC Design for WECS

System (28) can be used for controller design, bring (30) into (28), the drive train model can be rewritten as:(31)$$J_{t}\dot{\left(\frac{P_{g}}{T_{g}}\right)}=T_{a}-K_{t}\frac{P_{g}}{T_{g}}-T_{g}\;\left(\frac{\dot{P_{g}}T_{g}-P_{g}\dot{T_{g}}}{T_{g}{}^{2}}\right)=\frac{T_{a}}{J_{t}}-\frac{K_{t}}{J_{t}}\frac{P_{g}}{T_{g}}-\frac{1}{J_{t}}T_{g}\;\dot{P_{g}}T_{g}-P_{g}\dot{T_{g}}=\frac{1}{J_{t}}T_{a}T_{g}{}^{2}-\frac{K_{t}}{J_{t}}P_{g}T_{g}-\frac{1}{J_{t}}T_{g}{}^{3}$$

The modelling of wind energy conversion input-output control system is shown in (31), where the control input is the generator torque $T_{g}$ and the plant output is the generator power output $P_{g}$. Use u and y to replace $T_{g}$ and $P_{g}$ in (31):(32)$$\dot{y}u-y\dot{u}=\frac{1}{J_{t}}T_{a}u^{2}-\frac{K_{t}}{J_{t}}yu-\frac{1}{J_{t}}u^{3}$$

Based on U-model expression in [12], convert (32) into its U- realization:(33)$$\{\begin{array}{c} \dot{y}=\lambda_{0}f_{0}\left(\dot{u}\right)+\lambda_{1}f_{1}\left(\dot{u}\right)\\ \lambda_{0}=\frac{1}{J_{t}}T_{a}u-\frac{K_{t}}{J_{t}}y-\frac{1}{J_{t}}u^{2},\;f_{0}\left(\dot{u}\right)=1\\ \lambda_{1}=\frac{y}{u},\;f_{1}\left(\dot{u}\right)=\dot{u} \end{array}$$

**Remark** **7.**
*For inversion calculate of (32), accordingly,*

$\dot{u}=\frac{\dot{y}u}{y}-\frac{1}{J_{t}y}T_{a}u^{2}+\frac{K_{t}}{J_{t}}u+\frac{1}{J_{t}y}u^{3}$

*, then the control input is:*

$u=\int \dot{u}\;.\;$



**Remark** **8****.**
*In real-time wind energy control system, the value of the generator power output*

$P_{g}$

*will remain above 0 after the machine starts running, so the inversion is proper although it has y in the denominator.*


Based on the design procedures in Section 3.3, to assure a fast-tracking system without overshooting, this chapter chooses the damping ratio ζ =1 and undamped natural frequency $\omega_{n}=10$, the parameters in low-pass filter is: ρ=2 and λ=0.001, then the invariant controller is design by:(34)$$G_{c1}=\frac{G}{1-G}=\frac{1}{0.01s^{2}+0.2s}\;\mathrm{and}\;Q=\frac{1}{{\left(1+0.001s\right)}^{2}}$$

#### 4.1.3. Simulation Results

In this section, three-blade, horizontal-axis, and upwind variable speed wind turbine produced by WINDEY are selected as control plant for simulation experiments to demonstrate the efficiency of the proposed control method. The relevant parameters are given in Table 1.

The ratio between the desired power $P_{d}$ and the ideal maximum power $P_{amax}$ is $n_{p}=0.8$. Accordingly, the parameters in (33) can be calculated by:(35)$$\{\begin{array}{c} J_{t}=J_{r}+n_{g}^{2}J_{g}=5.7998\times 10^{6}\\ K_{t}=K_{r}+n_{g}^{2}K_{g}=4.4131\times 10^{3}\\ \frac{K_{t}}{J_{t}}=7.609\times 10^{-4} \end{array}$$


*Test 1: Tracking performance under uncertain wind speed and generator parameters*


The wind speed v is chosen to be the same in [25] with a mean of 9m/s and turbulence intensity of 10%. Because the speed measurement is inevitably inaccurate, this simulation experiment should consider the speed sensor noise. 

**Case** **1.**
*The speed measurement disturbance is chosen as a random value from −0.5 to 0.5 with sample time 0.1 s. Figure 7 shows the ideal wind speed for modelling and its real-time measurement trajectory.*


The simulation results for case 1 are shown in Figure 8. The generator output response and desired power output curves are shown in Figure 8a, both control methods obtain fine tracking performance in the presence of unknown measurable disturbance, where DOBUC method has smaller tracking error in Figure 8b. Figure 8c shows the disturbance observed result for system input compensation and Figure 8d shows the system input with little difference between the two control methods. When WECS only has unmeasurable wind speed disturbance, both control methods show the robustness for rejection. 

**Case** **2****.***The speed measurement disturbance is chosen as a random value from −0.5 to 0.5, meanwhile, modelling errors appear in the experiment with*$\Delta J_{t}=-0.2J_{t}$, $\Delta K_{t}=0.3K_{t}$*. Therefore, WECS (38) changed to:*(36)$$\dot{P_{g}}T_{g}-P_{g}\dot{T_{g}}=\frac{1.25}{J_{t}}T_{a}T_{g}{}^{2}-\frac{1.625K_{t}}{J_{t}}P_{g}T_{g}-\frac{1.25}{J_{t}}T_{g}{}^{3}$$

To demonstrate the superiority of the proposed control method further, the generator’s parameters are supposed to have variations from their nominal operation values. The simulation results for case 2 can be observed from Figure 9 that the proposed control method attain the better tracking performance. UDOB’s contribution is already very obvious from Figure 9c, and the tracking error under DOBUC method is therefore much smaller than U-control method from Figure 9b. When WECS only has both unmeasurable wind speed disturbance and modelling errors, the proposed UDOB demonstrates its efficiency for improving control system robustness. 


*Test 2: Tracking performance under uncertain wind speed, generator parameters and system input noise*


In real-time control application, the interference caused by noise or other signals in the environment to the system input of this WECS is unpreventable. Therefore, Case 3 introduces a disturbance to the input control signal channel, a sinusoidal signal with amplitude 0.3 (kN·m) and frequency 1 Hz to further test the robustness of the proposed control method.

The generator response results are shown in Figure 10. When the disturbance appears in the input channel of the control system, the tracking curve of U-control method has a large fluctuation, and DOBUC can perfectly track the desired power from Figure 10a. The proposed UDOB can accurately observe the input disturbance, and integrate all other errors including unmeasurable wind speed and system uncertainties into the input disturbance and then compensate from Figure 10c,d. Impartially, the proposed UDOB shows strong strength in disturbance observation and integration, and this input compensation control method for suppressing lumped disturbance is very convenient for the control system design and use.

### 4.2. Control of Permanent Magnet Synchronous Motors (PMSM) System

#### 4.2.1. Modelling of PMSM

Based on [29], consider system input disturbance, the mathematical model of PMSM in the rotating *d-q* reference frame can be described as the follows:(37)$$\{\begin{array}{l} \frac{d\omega_{r}}{dt}=\frac{3n_{p}\Phi_{v}}{2J}i_{q}+\frac{3n_{p}}{2J}\left(L_{d}-L_{q}\right)i_{d}i_{q}-\frac{B}{J}\omega_{r}-\frac{1}{J}T_{L}\\ \frac{di_{d}}{dt}=-\frac{R_{s}}{L_{d}}i_{d}+\frac{n_{p}L_{q}}{L_{d}}i_{q}\omega_{r}+\frac{1}{L_{d}}(u_{d}+d_{1})\\ \frac{di_{q}}{dt}=-\frac{R_{s}}{L_{d}}i_{q}-\frac{n_{p}L_{d}}{L_{q}}i_{d}\omega_{r}-\frac{n_{p}\Phi_{v}}{L_{q}}\omega_{r}+\frac{1}{L_{q}}(u_{q}+d_{2}) \end{array}$$ where $\omega_{r}$ the rotor speed, $i_{d}$,$\;i_{q}$ and $V_{d}$*,*$\;V_{q}$ are stator currents and voltages in *d-q* reference frame,

$L_{d}$ and $L_{q}$ are axes inductances in *d-q* reference frame, $T_{L}$ the load torque, $\Phi_{v}$ the rotor flux, J the inertia, $R_{s}$ the stator resistance, B the viscous friction coefficient and $n_{p}$ the number of pole pairs, $d_{1}$ and $d_{2}$ are control system input disturbances. 

The design aim is controlling voltages $V_{d}$ and $V_{q}$ to make rotor speed $\omega_{r}$ track a desired constant reference speed $\omega_{d}$ and the current $i_{d}$ is regulated to zero asymptotically. Let
(38)$$\{\begin{array}{l} x_{1}=\omega_{r},x_{2}=i_{d},x_{3}=i_{q},\;\;\;\;\;u_{1}=V_{d},\;u_{2}=V_{q}\\ a_{1}=\frac{3n_{p}\Phi_{v}}{2J},a_{2}=\frac{3n_{p}}{2J}\left(L_{d}-L_{q}\right),a_{3}=\frac{B}{J},a_{4}=\frac{1}{J}\;\;\;\;\\ b_{1}=\frac{R_{s}}{L_{d}}i_{d},b_{2}=\frac{n_{p}L_{q}}{L_{d}},b_{3}=\frac{1}{L_{d}}\;\\ c_{1}=\frac{R_{s}}{L_{d}},c_{2}=\frac{pL_{d}}{L_{q}},c_{3}=\frac{n_{p}\Phi_{v}}{L_{q}},c_{4}=\frac{1}{L_{q}} \end{array}$$

Consider system input disturbance, then system (38) can be rewritten into standard state space equation of:(39)$$\{\begin{array}{c} {\dot{x}}_{1}=a_{1}x_{3}+a_{2}x_{2}x_{3}-a_{3}x_{1}\;\;\;\;\;\;\;\;\;\;\;\;\;\;\;\\ {\dot{x}}_{2}=-b_{1}x_{2}+b_{2}x_{3}x_{1}+b_{3}u_{1}\;\;\;\;\;\;\;\;\;\;\;\;\\ {\dot{x}}_{3}=-c_{1}x_{3}-c_{2}x_{2}x_{1}-c_{3}x_{1}+c_{4}u_{2} \end{array}\;\mathrm{and}\;\{\begin{array}{c} y_{1}=x_{1}\\ y_{2}=x_{2} \end{array}$$

**Remark** **9.**
*It should be note that the lord torque*

$T_{L}$

*, system input disturbance*

$d_{1}$

*and*

$d_{2}$

*are unknow in (37) for modelling of PMSM, so they do not appear in (39).*


#### 4.2.2. DOBUC Design for PMSM

Assuming that the state variables in (39) are measurable. The invariant controller is same as (41) design. Based on U-model expression in [12], convert (39) into its U- realization:(40)$$\{\begin{array}{c} \dot{x}=\lambda_{0}f_{0}\left(u\right)+\lambda_{1}f_{1}\left(u\right)\\ y=h\left(x\right) \end{array}$$
where
$$\lambda_{0}=\left(\begin{array}{c} a_{1}x_{3}+a_{2}x_{2}x_{3}-a_{3}x_{1}\\ -b_{1}x_{2}+b_{2}x_{3}x_{1}\\ -c_{1}x_{3}-c_{2}x_{2}x_{1}-c_{3}x_{1} \end{array}\right),\;\lambda_{1}=\left(\begin{array}{cc} 0 & 0\\ b_{3} & 0\\ 0 & c_{4} \end{array}\right)\;f_{0}\left(u\right)=\left(1\right),\;f_{1}\left(u\right)=\left(\begin{array}{ccc} 0 & u_{1} & 0\\ 0 & 0 & u_{2} \end{array}\right),\;h\left(x\right)=\left(\begin{array}{c} x_{1}\\ x_{2} \end{array}\right)$$

Based on the design procedures in Section 3.3, to assure a fast-tracking system without overshooting, this chapter chooses the damping ratio =1 and undamped natural frequency $\omega_{n}=10$, the parameters in low-pass filter is: ρ=2 and λ=0.001, then the invariant controller is design by:(41)$$G_{c1}=\frac{G}{1-G}=\frac{1}{0.01s^{2}+0.2s}\;\mathrm{and}\;Q=\frac{1}{{\left(1+0.001s\right)}^{2}}$$

#### 4.2.3. NDOBC Design for PMSM

Assuming that the state variables in (39) are measurable. Ref. [30] designed a nonlinear disturbance observer (NDO) based robust tracking controller, which can compensate for modelling errors and system input disturbance of PMSM system is given by: (42)$$u=A^{-1}\left(x\right)\left[-B\left(x\right)+V\left(x,y_{r}\right)+T\left(x\right)\hat{d}\right]$$
where $\hat{d}$ is the estimate value of all the disturbances and uncertainties in Section 3.3. The other parameters in (42) are:$$A=\left[\begin{array}{cc} 0 & \frac{K_{t}}{JL_{q}}\\ \frac{1}{L_{d}} & 0 \end{array}\right],\;B=\left[\begin{array}{c} \frac{K_{t}}{J}\left(-\frac{R_{s}}{L_{q}}i_{q}-n_{p}\omega_{r}i_{d}-\frac{n_{p}\phi }{L_{q}}\omega \right)-\frac{B}{J}\left(\frac{K_{t}}{J}i_{q}-\frac{B}{J}\omega_{r}\right)\\ -\frac{R_{s}}{L_{q}}i_{d}+n_{p}\omega_{r}i_{q} \end{array}\right],\;V=\left[\begin{array}{c} -C_{1}\left(\omega_{r}-\omega_{d}\right)-C_{2}\left(\frac{K_{t}}{J}i_{q}-\frac{B}{J}\omega_{r}\right)\\ -C_{3}\left(i_{d}-0\right) \end{array}\right],\;T=\left[\begin{array}{ccc} -C_{2}+\frac{B}{J} & 0 & -\frac{K_{t}}{J}\\ 0 & -1 & 0 \end{array}\right]\;with\;K_{t}=\frac{3n_{p}\Phi_{v}}{2J}$$

The specific design process and analysis can refer to [30].

#### 4.2.4. Simulation Results

To demonstrate the efficiency of the proposed DOBUC method, simulation experiments of controlling the PMSM system under DOBUC and NDOBC are carried out in this chapter. The parameters of the PMSM are the same as [30]: $n_{p}=4$, $R_{s}=1.74Ω$, $\Phi_{v}=0.1167$ wb, $L_{d}=L_{q}=0.004$ H, $B=7.403\times 10^{-5}$ N·m·s.rad, $J=1.74\times 10^{-4}$ Kg·$m^{2}$, reference rate speed $\omega_{d}=3000$ rpm. The control parameters in (41) are:$$C_{1}=8000,C_{2}=180,C_{3}=300$$

And the gain matrix for NDOB design is: $$l=\left[\begin{array}{ccc} 500 & 0 & 0\\ 0 & 500 & 0\\ 0 & 0 & 500 \end{array}\right]$$


*Test 1: Tracking performance under unknown load torque*


The unknown load torque $T_{L}=2$ N·m is added to the PMSM from t=0.2 s to 0.6 s. The desired rotor speed $\omega_{d}=3000\;\mathrm{rpm}=100\pi \;\mathrm{rad}/s$. The simulation results for Test 1 are shown in Figure 11. As shown in Figure 11a, the proposed DOBUC method obtains fine tracking performance but NDOBC method is unable to suppress the overshoot caused by load torque. The output of disturbance observer in NDOBC and DOBUC systems are shown in Figure 11c,d, respectively.


*Test 2: Tracking performance under control system input disturbance*


In real-time control application, the system input disturbance caused by noise or other environmental signals of PMSM is unpreventable. Therefore, test 2 introduces a disturbance to the input control signal channel, a sinusoidal signal with amplitude 1 voltage and frequency 3 Hz. Both control methods can suppress control input disturbance from Figure 12a, DOBUC get better control performance than NDOBC. From Figure 12c,d, both disturbance observers can accurately estimate the input disturbance.


*Test 3: Tracking performance under system input disturbance, uncertain PMSM parameters and lord torque*


To further demonstrate the robustness of the proposed control method, the electrical parameters in the modelling are supposed to be inaccurate and result in variations from their nominal values. The following parameters have perturbations: $\Delta R_{s}=-0.2R_{s}$, $\Delta L_{q}=0.3L_{q}$, $\Delta L_{d}=-0.3L_{d}$, J=0.4J, and B=−0.5B. The other disturbances are the same as previous tests.

The rotor response results are shown in Figure 13. From Figure 13a,b, NDOBC method can enforce PMSM to be stable, but it cannot suppress the tracking error caused by disturbances. It also can observer that the proposed DOBUC method can attain good tracking requirements. The output of disturbance observer in NDOBC and DOBUC systems are shown in Figure 13c,d, respectively.

### 4.3. Discussion

The experiments carried out above demonstrate that DOBUC method has strong robustness, fast response, and good tracking ability. From WECS control tests, compared with the traditional U-control, DOBUC undoubtedly has a stronger disturbance rejection ability. By introducing UDOB (Combines linear frequency-domain DOBC structure and UM dynamic inversion), U-control system can improve its robustness through system input error compensation.

From PMSM control tests, compared with the NDOBC, DOBUC shows great potential in convenient design. Experiments results show that all disturbance can be observed regardless of whether it is NDOB or UDOB. The difference is that the improved UDOB integrates the lumped disturbance into the input channel and NDOB observes the overall disturbance of each state variable. However, the control performance of NDOBC is limited by the baseline feedback controller. On the contrary, the good tracking performance of DOBUC further proves that it is feasible and reasonable to extend linear frequency-domain DOB to adapt to nonlinearity.

## 5. Conclusions

This paper presents a novel disturbance-observer-based U-control method inspired by the frequency-domain DOBC structure. This novel DOB applies to linear and nonlinear control systems with invertible controlled plants/processes. With the demonstration of the WECS and PMSM simulation tests, the U-control method has considered the full complex nonlinear dynamics, and the proposed DOB can observer all system input disturbance and modelling uncertainties. Different from the time-domain NDOBC, the proposed UDOB can integrate all lumped disturbances into the control input channel and perform compensation design, which provides convenience and new ideas for the researches of robust control systems combined with other mature control methods.

However, subject to the characteristics of UM dynamic inversion algorithm, this proposed DOB can only be applied to a type of invertible and bounded plants/processes. For systems that are unbounded or complex inversion (large computation), the performance of the proposed DOB will be critically restricted. In summary, for systems with a higher order that meet the mentioned assumptions, it is more convenient to use DOBUC; for systems that do not meet the assumptions or the inversion process requires large computation, the classic NDOBC is more efficient. At the same time, integrating all lumped disturbances into the control input channel also means that the allocation of input disturbance compensation will become a challenge for the multiple input multiple output (MIMO) systems, how to efficiently allocate disturbance compensation in MIMO systems will be the next research direction.

## Figures and Tables

**Figure 1 entropy-23-01625-f001:**
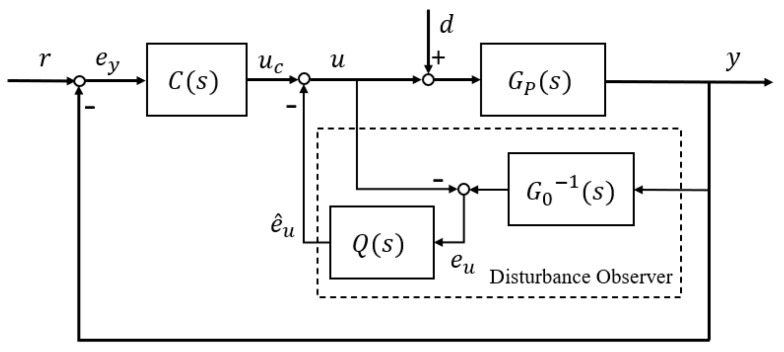
Conceptual diagram of frequency-domain disturbance observer based control DOBC.

**Figure 2 entropy-23-01625-f002:**
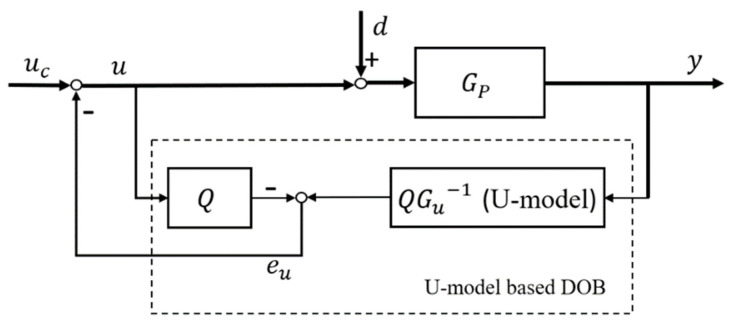
Conceptual structure of UDOB.

**Figure 3 entropy-23-01625-f003:**
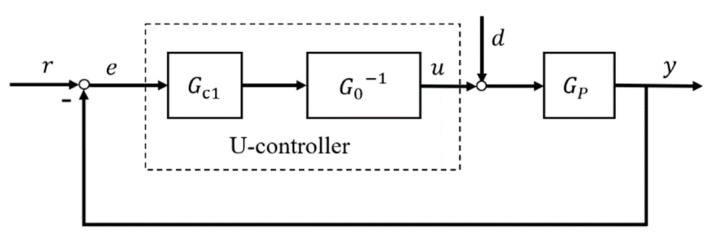
U-control system design framework.

**Figure 4 entropy-23-01625-f004:**
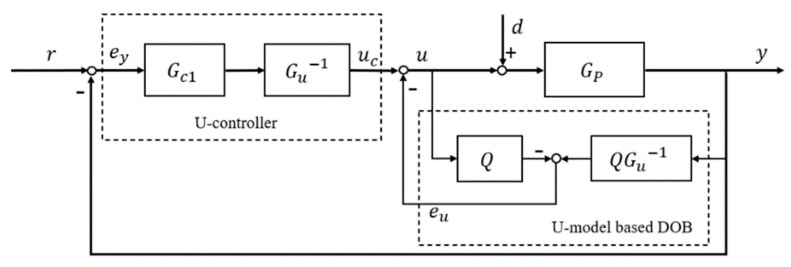
Conceptual structure of disturbance observer based U-control DOBUC.

**Figure 5 entropy-23-01625-f005:**
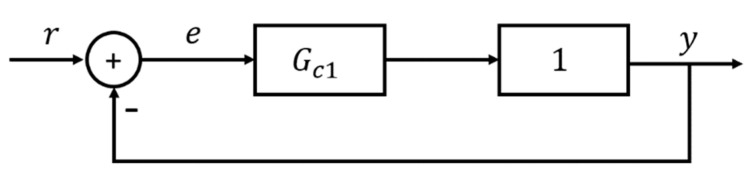
Equivalent structure after UDOB compensation.

**Figure 6 entropy-23-01625-f006:**
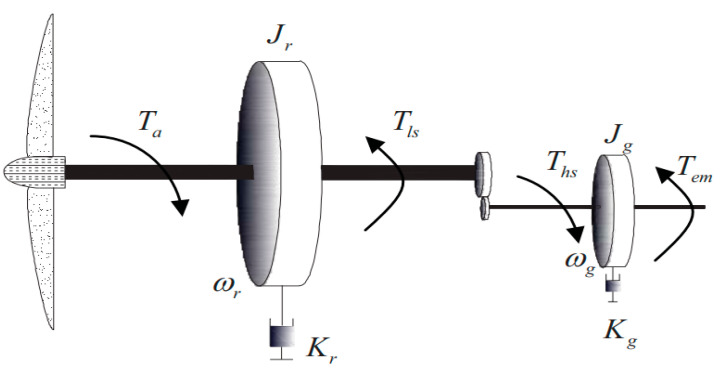
Schematic diagram of drive train.

**Figure 7 entropy-23-01625-f007:**
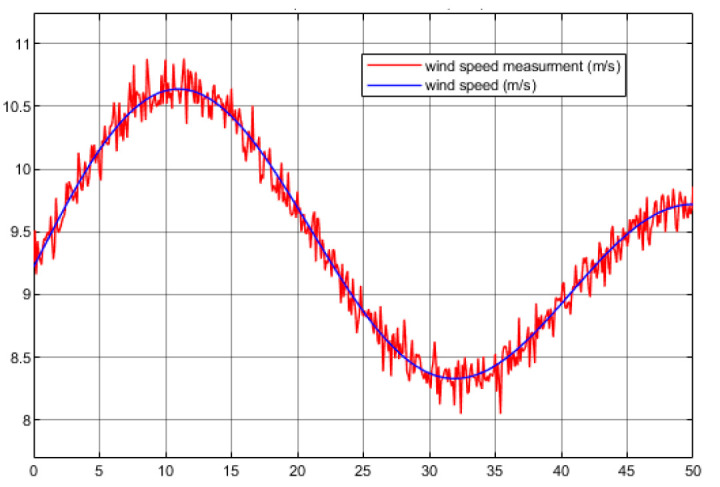
Ideal wind speed with its measurement.

**Figure 8 entropy-23-01625-f008:**
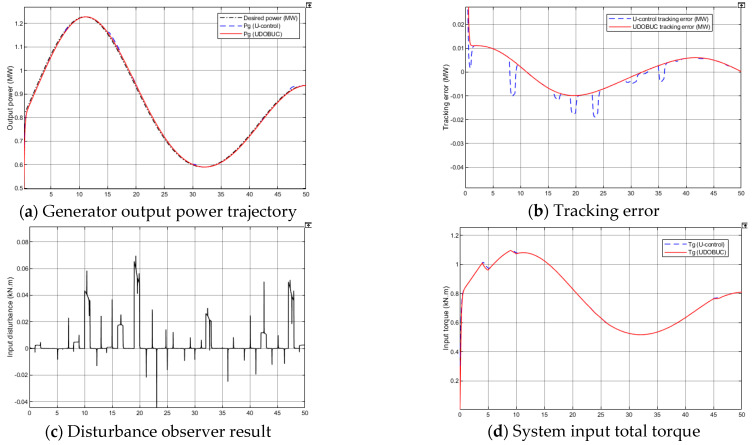
Simulation results of WECS case 1 in Test 1.

**Figure 9 entropy-23-01625-f009:**
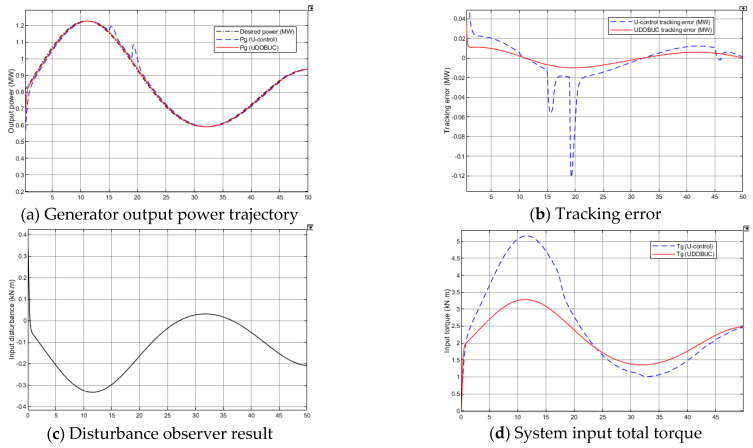
Simulation results of WECS case 2 in Test 1.

**Figure 10 entropy-23-01625-f010:**
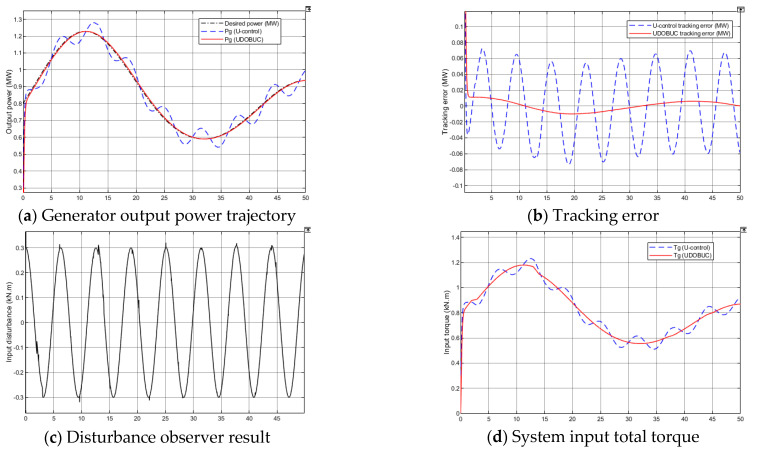
Simulation results in Test 2.

**Figure 11 entropy-23-01625-f011:**
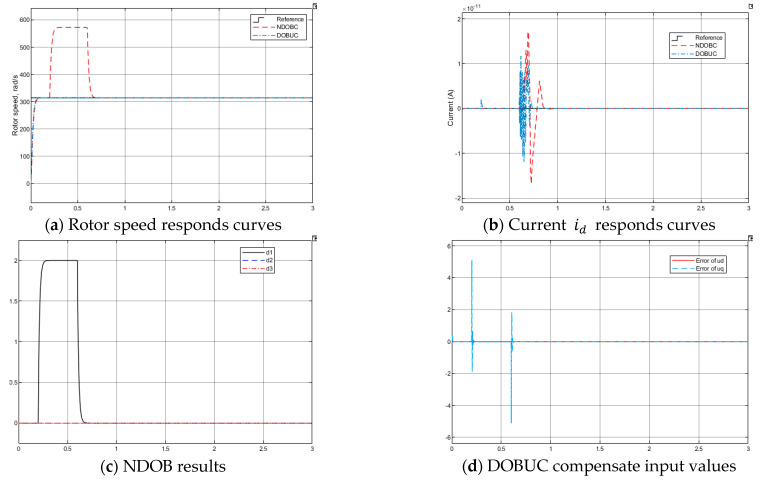
Simulation results of PMSM in Test 1.

**Figure 12 entropy-23-01625-f012:**
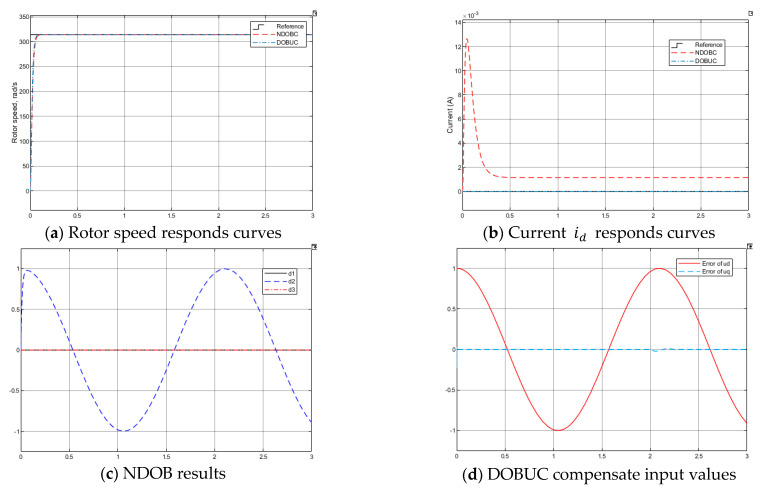
Simulation results of PMSM in Test 2.

**Figure 13 entropy-23-01625-f013:**
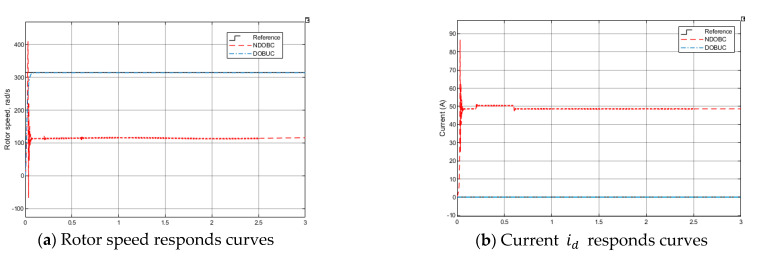
Simulation results of PMSM in Test 3.

**Table 1 entropy-23-01625-t001:** Wind turbine parameters.

Rated power	1.5 MW
Rotor radius	*R* = 38.5 m
Rotor inertia	*J_r_* = 4,456,761 kg·m^2^
Generator inertia	*J_g_* = 123 kg·m^2^
Rotor friction coefficient	*K_r_* = 45.52 N·m/rad/s
Generator friction coefficient	*K_g_* = 0.4 N·m/rad/s
Gearbox ratio	*n_g_* = 104.494
Air density	ρ=1.12

## Data Availability

Project files about DOBUC simulation experiments are available at https://pan.baidu.com/s/1Vzjve7e6i2EXnygO7tTPqg With Code: x4jh.

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
