# Peer review of "Disturbance-Observer-Based U-Control (DOBUC) for Nonlinear Dynamic Systems"

_entropy, 2021, doi:10.3390/e23121625_

Round 1
Reviewer 1 Report
This paper deals with the problem of disturbance observer based U-control (DOBUC) for nonlinear dynamic systems, and the proposed method is verified by simulation results. However, there are some problems in the paper as follows.
(1) In Section 1, the authors should discuss why they design disturbance observer in frequency domain, not time domain. In addition, the contribution and novelty of the paper should be further clarified.
(2) Section 2 is too long and can be divided into several sections.
(3) In Lines 104 and 105, the numbers are wrong.
(4) In Line 189, the sentence “the design parameters in NDOB design are more complex…” is inaccurate. The design parameters of NDOB can be simplified.
(5) In Line 199, the number is wrong.
(6) In Section 2.2.3, the key parameters of the observer and controller should be discussed.
(7) In simulation, the proposed method should be compared with advanced ones with superior performance.
(8) The writing of the paper needs to be polished.
Author Response
Comments and Suggestions for Authors
This paper deals with the problem of disturbance observer based U-control (DOBUC) for nonlinear dynamic systems, and the proposed method is verified by simulation results. However, there are some problems in the paper as follows.
- In Section 1, the authors should discuss why they design disturbance observer in frequency domain, not time domain. In addition, the contribution and novelty of the paper should be further clarified.
The fundamental idea of the DOB was first proposed in terms of the frequency domain, which aims bring together all the uncertainties and disturbances as a single lumped disturbance term and then estimate it. Due to the limited estimated performance in frequency domain DOB (requiring linear systems), time-domain DOB methods have been developed and widely implemented to many practical applications. It should be noted that the estimated disturbance from time domain DOB cannot be used to compensated directly, because the disturbances and control inputs are not in the same channels, where but . Therefore, finding a way to extend conventional linear system based frequency-domain DOB to be applicable to nonlinear systems, is still an effective way to design nonlinear DOB.
The contributions of this paper are revised in section 1. Please refer to the revised/resubmitted paper for the clarification/explanation.
- Section 2 is too long and can be divided into several sections.
Thanks for your suggestion. Section 2 has been re-organized and divided into 2 sections. Please refer to the paper for the re-structured.
- In Lines 104 and 105, the numbers are wrong.
Yes, the numbers have been corrected.
- In Line 189, the sentence “the design parameters in NDOB design are more complex…” is inaccurate. The design parameters of NDOB can be simplified.
Yes, it is an inaccurate statement. It should be: ”the design in NDOB is more complex”. In frequency-domain DOB, the only designing object is the low-pass filter. Therefore, compare with frequency-domain DOB, the design procedure of time-domain NDOB is more complicated, as well as the observer parameters computation.
- In Line 199, the number is wrong.
Yes, the numbers have been corrected.
- In Section 2.2.3, the key parameters of the observer and controller should be discussed.
Yes. The parameters chosen are discussed more specifically at each design step in the proposed control procedure. In section 4, how to design the parameters based on the desired control performance is also discussed. Please refer to the revised paper for the details.
- In simulation, the proposed method should be compared with advanced ones with superior performance.
Yes. In section 4.1, the proposed method is compared with U-control method in Control of Wind energy conversion system. In section 4.2, the proposed method is compared with time-domain NDOBC in control of Permanent Magnet Synchronous Motors (PMSM) System. Both simulation experiments consider the system external disturbances and modelling uncertainties, and the simulation results show the superior control performance of proposed control method over the compared ones.
- The writing of the paper needs to be polished.
Yes, with the author’s best effort. Please help to check through the revised draft.
Reviewer 2 Report
The novelty of the paper is not clear. Authors should also improve the quality of all figures.
Author Response
Comments and Suggestions for Authors
The novelty of the paper is not clear. Authors should also improve the quality of all figures.
The contributions and novelty of this paper are re-written in section 1. Please check through the revised/resubmitted draft for the clarification/explanation.
The quality of all figures has been improved. Please refer to the paper for details.
Round 2
Reviewer 1 Report
The paper has been modified according to the review comments in last round and can be accepted for publication in Entropy in present form.
Reviewer 2 Report
The authors have addressed all my comments.